# Potato (*Solanum tuberosum*) as a New Host for *Pentastiridius leporinus* (Hemiptera: Cixiidae) and *Candidatus* Arsenophonus Phytopathogenicus

**DOI:** 10.3390/insects14030281

**Published:** 2023-03-13

**Authors:** Sarah Christin Behrmann, André Rinklef, Christian Lang, Andreas Vilcinskas, Kwang-Zin Lee

**Affiliations:** 1Institute for Insect Biotechnology, Justus Liebig University of Giessen, Heinrich-Buff-Ring 26, D-35392 Giessen, Germany; 2Fraunhofer Institute for Molecular Biology and Applied Ecology, Ohlebergsweg 12, D-35394 Giessen, Germany; 3Association of Hessian-Palatinate Sugar Beet Growers e.V., Rathenaustrasse 10, D-67547 Worms, Germany

**Keywords:** Cixiidae, host shift, bivoltine, Solanaceae, Hemiptera, proteobacteria, life cycle, field data, phylogeny, cytochrome oxidase I

## Abstract

**Simple Summary:**

*Pentastiridius leporinus* is a planthopper that transmits two bacterial pathogens to sugar beet plants: Arsenophonus and Stolbur. These bacteria cause an economically important disease known as syndrome basses richesses (SBR), characterized by deformed yellow leaves and low beet yields. Having observed potato fields in Germany infested with planthoppers and showing signs of leaf yellowing, we set out to identify the planthoppers and any transmitted pathogens. We identified *P. leporinus* by comparisons of COI and COII partial sequences and detected both bacterial pathogens in planthoppers, as well as in beet and potato plants. We found that *P. leporinus* nymphs and adults can transmit the bacteria and that long, warm summers allow two generations of planthoppers to breed, probably increasing the population size and threat of disease transmission in the following year. We conclude that the planthopper *P. leporinus* has expanded its host range to potato, and that further studies are needed, to facilitate the development of control strategies that minimize losses in sugar beet and potato crops.

**Abstract:**

*Pentastiridius leporinus* is a planthopper (Hemiptera: Cixiidae) that vectors two phloem-restricted bacterial pathogens to sugar beet (*Beta vulgaris* (L.)): the γ-proteobacterium *Candidatus* Arsenophonus phytopathogenicus and the stolbur phytoplasma *Candidatus Phytoplasma solani*. These bacteria cause an economically important disease known as syndrome basses richesses (SBR), characterized by yellowing, deformed leaves and low beet yields. Having observed potato fields in Germany infested with cixiid planthoppers and showing signs of leaf yellowing, we used morphological criteria and COI and COII as molecular markers, to identify the planthoppers (adults and nymphs) primarily as *P. leporinus*. We analyzed planthoppers, potato tubers, and sugar beet roots and detected both pathogens in all sample types, confirming that *P. leporinus* adults and nymphs can transmit the bacteria. This is the first time that *P. leporinus* has been shown to transmit Arsenophonus to potato plants. We also found that two generations of *P. leporinus* were produced in the warm summer of 2022, which will probably increase the pest population size (and thus the prevalence of SBR) in 2023. We conclude that *P. leporinus* has expanded its host range to potato, and can now utilize both host plants during its developmental cycle, a finding that will facilitate the development of more efficient control strategies.

## 1. Introduction

The planthopper *Pentastiridius leporinus* (Linnaeus) belongs to the family Cixiidae (Hemiptera: Fulgoromorpha), and has been known to colonize sugar beet plants (*Beta vulgaris* L.) since the early 2000s [1,2,3]. Before this, the adults were known to feed on reed grass (*Phragmites australis* (Cav.)), in moist or wet sites along rivers or in ruderal and brackish ecosystems, but the host plant for the nymphs was unknown [4]. The recent host shift has enabled *P. leporinus* to complete its life cycle on sugar beet (adult feeding, oviposition, hatching, and nymphal feeding). After the harvesting of sugar beet in autumn, nymphs overwinter on subsequent crops, typically winter wheat (*Triticum aestivum* (L.)) or barley (*Hordeum vulgare* (L.)) [5]. The adult planthoppers begin to emerge the following May and show flight activity usually until August, and sometimes until September if the summer is hot and dry [6]. Oviposition takes place at the beginning of July. Laboratory assays showed that the time from the first larval stage until adult emergence, under controlled conditions, is >100 days, without an induced diapause [6].

*Pentastiridius leporinus* is the main insect vector of the bacterial disease of sugar beet known as syndrome basses richesses (SBR), in Germany and Switzerland [7,8,9]. Symptoms of SBR include the yellowing of older leaves, lancet-shaped leaf deformations, and necrosis of the vascular tissue, which reduces the sugar content of the beet roots [10]. The syndrome is associated with two different bacterial pathogens: the γ-proteobacterium *Candidatus* Arsenophonus phytopathogenicus (hereafter described as Arsenophonus) and the stolbur phytoplasma *Candidatus Phytoplasma solani* (hereafter described as stolbur) [10]. Arsenophonus has been more abundant in sugar beet over the last five years, based on monitoring activities in Germany [7,11].

Bacteria of the clade Arsenophonus typically associate with insects as mutualists or parasites, but they are also pathogens of sugar beet and strawberry, where they cause marginal chlorosis disease [5]. Arsenophonus is transmitted by both adults and nymphs of *P. leporinus* [10,12]. An inoculation access period (IAP) of three days was tested in adults, revealing that females lived longer and were better vectors than males, with 64% and 24% efficiency of transmission, respectively [10]. Nymphs were hosted for 30 days on sugar beet seedlings, and Arsenophonus transmission was successful in four of six replicates [12]. The bacteria were also vertically transmitted to the offspring [12]. Arsenophonus was abundant in most internal organs of *P. leporinus*, with the highest bacterial load in the salivary glands and gonads of adults [12].

Stolbur is responsible for rubbery tap root disease (RTD) in sugar beet in Southeast Europe [13], and was first observed in Serbia in the 1950s [14]. Stolbur is also known to cause bois noir in grapevines (*Vitis vinifera* L.) [15]. This is transmitted by *Hyalesthes obsoletus* (Signoret), another cixiid planthopper known for its ability to complete its life cycle on different host plants, such as the nettle *Urtica dioica* (L.) and bindweed *Convolvus arvensis* (L.) [16]. Several other crops are damaged by *H. obsoletus*, including celery, lavender, maize, and salvia [17,18,19,20]. Recently, we detected a high abundance of stolbur 16sR-XII in *P. leporinus* captured in sugar beet fields in Rhineland-Palatinate and South Hesse [6]. The same insects were also carrying Arsenophonus.

In potato (*Solanum tuberosum* (L.)), stolbur is known to cause diverse symptoms including reddening, aerial tubers, leaf discoloration, shortened internodes, and upward rolling of the top leaf, resulting in severe yield losses [21]. The identified main vector insect of stolbur in potato was *Hyalesthes obsoletus* [22]. *P. leporinus* had neither been found in potato nor transmitting stolbur. The detection of stolbur in sugar beet in 2020 and 2021 drew our attention to nearby potato fields, that started to show yellowing symptoms. The aim of this study was to identify the planthopper associated with wilting potato plants, and to characterize the pathogens responsible for this novel syndrome.

## 2. Materials and Methods

### 2.1. Field Collection of P. leporinus from Potato Tubers and Sugar Beet Roots

*P. leporinus* samples were collected from three locations (Table 1) on neighboring sugar beet and potato fields (<300 m apart), in September 2022. Adults were collected using a sweep net. Nymphs were collected by digging them out from between potato tubers, or by using a beet fork to collect them from along the beet roots. Field-collected nymphs were transferred individually to microcentrifuge tubes or Petri dishes for further analysis in the laboratory. Field-collected adults were transferred to a 2-week-old potted potato plant var. Jule (Solana, Windeby, Germany), for egg deposition and hatching of first-instar nymphs. Plants were kept in the greenhouse in single cages (Cavea PopUp size M net cage (Howitec, Bolsward, The Netherlands)). The temperature of the chamber was set to 22/16 °C day/night, with a 16 h photoperiod. Images of field-collected and freshly-hatched nymphs were captured using a VHX-5000 digital microscope (Keyence Deutschland, Neu-Isenburg, Germany), and measured using the VHX-5000 communication software.

Potato tubers expressing stolbur-like symptoms, characterized by the formation of side shoots, wilting, and rubbery tubers, were collected from the same locations. In the Eich and Ibersheim locations, we also collected 15 sugar beet samples with SBR symptoms, characterized by yellowing leaves, deformed young leaves, and necrosis of the vascular tissue and root tips.

### 2.2. Identification of Nymphs and Adults

To confirm the species, adults were morphologically examined [23] and DNA was extracted from nymphs and adults representing each location using the DNeasy Blood & Tissue Kit (Qiagen, Hilden, Germany). A single fragment of the partial cytochrome oxidase subunit I and II (COI and COII) genes was amplified, as previously described [8], using primer pair C1J2441/C2N3661 [24]. The amplicons were purified with Exonuclease and SAP (New England Biolabs, Ipswich, MA, USA), followed by sequencing (Eurofins Genomics, Cologne, Germany). In total, seventeen adults were analyzed: for potato, four from Eich, seven from Lampertheim and five from Ibersheim; and for sugar beet, one from Ibersheim. As for the nymphs, nine were analyzed, one from potato in Eich, two from potato in Ibersheim, four from potato in Lampertheim, one from sugar beet in Eich, and one from sugar beet in Ibersheim. COI/COII sequences from three *P. leporinus* nymphs, collected from potato fields, one adult, collected from a sugar beet field, and reference sequences FN179288–FN179291 from *P. leporinus* and *Hyalesthes obsoletus*, were compared [8]. COI and COII sequences were trimmed to 972 bp and aligned using MUSCLE [25], and a phylogenetic tree was created using the neighbor-joining method and Geneious v10.2 (Biomatters, Auckland, New Zealand). Bootstrap analysis was conducted using 10,000 replications. The COI/COII sequences were submitted to Genbank, under accession numbers with sequential numbering from OQ420381 to OQ420405.

### 2.3. Abundance of P. leporinus in 2022

The abundance of *P. leporinus* was evaluated using transparent, sticky “cloak” traps (PAL, CSALAMON Group, Budapest, Hungary), 25 × 35 cm. The traps were set up at one location in Hesse (coordinates 49.758758, 8.577285). The site was a commercial sugar beet field, 13.2 km away from the examined potato field in Eich (Table 1). Three traps were placed at distances of 10, 30, and 50 m from the edge of the field, on a pole at a height of 10 cm above the sugar beet vegetation. The sampling period started in May and ended in September, after two continuous weeks without catches. The traps were replaced weekly and the number of adults was determined. To evaluate the ability of *P. leporinus* nymphs to develop on potato plants, the head capsule width and dorsal length of 100 field-collected nymphs were measured, using a VHX-5000 digital microscope.

### 2.4. Transmission Assays with Nymphs

The ability of *P. leporinus* nymphs to transmit Arsenophonus and stolbur to uninfected potato plants, was tested by releasing five field-collected nymphs (third to fifth instar) onto twelve potato plants and three sugar beet plants, followed by feeding for an IAP of 66 days. The control group consisted of three potato plants without nymphs. Field-collected *P. leporinus* nymphs were placed directly onto 3-week-old sugar beet plants (KWS Annarosa, Einbeck, Germany) and 5-day-old potato plants (Colomba HZPC, Joure, The Netherlands), growing in Göttinger 3 L pots (Lamprecht-Verpackungen, Göttingen, Germany), filled with Frühstorfer potting soil LD 80 (Heinrichs, Ingelheim, Germany) and covered with expanded clay (Floragard, Oldenburg, Germany). Plants were kept in single cage, in a Cavea PopUp size M net cage (Howitec, Bolsward, The Netherlands), and watered moderately twice weekly. The temperature in the greenhouse chamber was set to 22/16 °C day/night, with a 16 h photoperiod.

After 66 days, nymphs were collected from plants and pooled by plant origin for DNA extraction, as described above. Potato plants were divided into leaves, upper stem, midsection, lower stem, tubers, and roots. Sugar beet plants were divided into leaves, leaf stems, upper root, midsection of root, root tip, and tender roots. DNA extracts from plants were prepared from 0.1 g samples according to a modified CTAB procedure [26], followed by qRT-PCR (Section 2.6).

### 2.5. Comparison of Arsenophonus Isolated from Potato Tubers, Beet Roots, and P. leporinus

Nested PCR with primer pairs Fra5/rP1 and Alb1/Oliv1, was carried out to amplify the Arsenophonus 16S ITS sequence [3]. The products were separated by 2% agarose gel electrophoresis and stained with 0.1 µL/ml SYBR-Safe (Thermo Fisher Scientific, Waltham, MA, USA), followed by visualization under UV light. The products were then cleaned up and sequenced as described above. Multiple sequences were aligned with MUSCLE, using Geneious v10.2, as described above. The Arsenophonus sequences were submitted to Genbank, under accession numbers with sequential numbering from OQ411083 to OQ411090.

### 2.6. Analysis of Arsenophonus and Stolbur Infections

Ninety-one individual insects, from three locations, were analyzed: thirty-six adults and thirty-six nymphs from potato fields, as well as nine adults and ten nymphs from sugar beet fields. Following DNA extraction, as described above, TaqMan qRT-PCR was performed using Luna Universal qRT-PCR Master Mix (New England Biolabs, Frankfurt, Germany), 25 ng of template DNA, and primer pairs KL437/438 and KL464/465, for the detection of Arsenophonus and stolbur, respectively [7]. The number of target copies was determined using absolute standard curves. Samples with a CT value > 35 were considered negative, which is equivalent to 22 copies for Arsenophonus and 14 copies for stolbur.

To assess the abundance of infected potato and sugar beet plants in two locations, 30 symptomatic sugar beet root samples and 45 symptomatic potato tuber samples were collected, from three locations, and tested for Arsenophonus and stolbur. DNA extracts were prepared from 0.1 g tuber or sugar beet root samples, using a modified CTAB method [26] followed by qRT-PCR, as described above.

## 3. Results

### 3.1. Identification and Morphometric Analysis of P. leporinus on Potato Plants

A recent increase in wilting syndromes in potato fields in Southwest Germany, was accompanied by the observation of hitherto unknown planthoppers in those affected fields. We identified the species as *P. leporinus,* based on morphological traits in potatoes (*Solanum tuberosum* var. Juventa). We found symptomatic potatoes colonized by nymphs, filamentous residues reminiscent of the characteristic wax structures of egg clutches, and hatching *P. leporinus* adults (Figure 1).

Morphometric analysis was carried out on 100 of the field-collected nymphs (Appendix A).

### 3.2. Molecular Identification of P. leporinus

We analyzed 23 individual nymphs and adults from potato and sugar beet fields, all of which (except one nymph) were identified as *P. leporinus*. The one exception, collected from sugar beet, was identified as *Cixius* sp., another planthopper in the family Cixiidae. Evolutionary relationships among the planthoppers and some reference specimens from other years and locations, were checked on the basis of COI and COII sequence similarity, resulting in the phylogenic tree shown in Figure 2.

Based on COI and COII sequences, the planthoppers found in potato and sugar beet showed a closer genetic relationship to *P. leporinus* than the two known references from France and Russia. The nearest related species (*P. beieri*), as well as the more distantly related *H. obsoletus* and *Cixius* sp. specimens, showed clear sequence differences.

### 3.3. Second Generation of Adults in Bickenbach, 2022

In 2022, 1336 *P. leporinus* adults were caught in a sugar beet field in Bickenbach, Hesse using transparent, sticky traps (Figure 3). The flight period lasted from calendar week 20 (mid-May) until calendar week 39 (end of September). At this location, 92.4% of the captured individuals were males.

Surprisingly, two peaks of abundance were observed. The first occurred at the beginning of July (calendar week 27), followed by a decline until the end of July (calendar week 30), at which point the mean capture rate per trap fell to a low of 0.3 insects. Afterwards, the number of captured adults rose again, reaching a second peak at the end of August (calendar week 35). One insect was trapped at the end of September (calendar week 39). 

### 3.4. Transmission Assay

The detection of Arsenophonus and stolbur in nymphs and plants, by qRT-PCR, is summarized in Table 2. All groups of nymphs tested positive for Arsenophonus, but only one of fifteen groups tested positive for stolbur. Three of twelve potato plants tested positive for Arsenophonus, whereas stolbur was not detected in any potato plants. In the first potato plant, Arsenophonus was detected in the tubers with 45 copies. In the second plant, 141 copies were found in the roots. The third potato plant was more severely infected, with 85,635 copies in the lower stem, 5341 in the roots, 172 copies in the leaves, and 93 in the tubers. Arsenophonus and stolbur were both detected in one of the three sugar beet plants, which contained 23 copies of Arsenophonus and 1416 copies of stolbur in the root tip. Neither pathogen was found in the control group. 

### 3.5. Similarity of Arsenophonus in All Organisms

The Arsenophonus qRT-PCR results were verified by nested PCR followed by agarose gel electrophoresis [3]. Infected potato tubers showed the same characteristic banding pattern as infected sugar beet and planthopper samples (Appendix A). No bands were observed in the negative control. Two Arsenophonus amplicons from planthoppers, one from sugar beet, and four from potato plants were sequenced, revealing 100% identity with the Arsenophonus 16S ITS sequence DQ834353.1 (positions 80–419 bp) submitted by [3].

### 3.6. Prevalence of Arsenophonus and Stolbur in P. leporinus and Plant Material

We tested 91 *P. leporinus* adults and nymphs for Arsenophonus and stolbur, 69.2% of which were positive for Arsenophonus and 2.2% of which were positive for stolbur. Each adult positive for stolbur was also positive for Arsenophonus (Figure 4).

The prevalence of Arsenophonus in both plant species was high: 95% of the sugar beet roots and 100% of the potato tuber samples were positive for this species. In addition, stolbur was detected in 20% of the potato tubers and 45% of the beet root samples (Figure 5). 

We analyzed forty-five potato tuber samples in total. Among the fifteen samples from Eich, all were positive only for Arsenophonus, except a single uninfected sample. Among the fifteen samples from Ibersheim, seven were positive only for Arsenophonus, seven were positive for both pathogens and one was positive only for stolbur. Among the fifteen samples from Lampertheim, fourteen were positive for Arsenophonus alone and one sample was positive for both pathogens. We analyzed thirty sugar beet samples in total, all of which were positive for Arsenophonus or a double infection. Among the fifteen samples from Eich, seven were positive for Arsenophonus and eight were positive for both pathogens. Among the fifteen samples from Ibersheim, nine were positive for Arsenophonus and six were positive for both pathogens. 

## 4. Discussion

Several species of planthoppers and psyllids cause serious damage to potato crops by direct feeding and/or by transmitting pathogens, especially phytoplasmas, bacteria, and viruses [27]. Many planthopper species associated with potato fields belong to the family Cixiidae, including *H. obsoletus*, *Reptalus quinquecostatus* (Dufour) and *R. panzeri* (Löw) [22]. Our study presents the first molecular evidence that potato plants are colonized by *P. leporinus*. We also confirmed adult feeding, egg laying, hatching, and nymphal feeding on potato plants and tubers. Our data therefore suggest that *P. leporinus* can move between sugar beet and potato host plants, as represented in an updated life cycle diagram (Figure 6). To compare the suitability of sugar beet and potato as hosts, long-term life cycle trials are needed to understand the dynamics of *P. leporinus* developmental biology. To compare these host interactions, suitable protocols for some host plants under controlled conditions have already been established by [6].

The identification of species was not possible based on morphological characteristics alone during the nymphal stages, so we sequenced COI amplicons to provide species-dependent molecular markers. We found that the nymphs and adults on sugar beet and potato plants were 100% identical to the *P. leporinus* reference sequences, which differed from the *H. obsoletus* and *Cixius* sp. amplicons. However, further genomic analysis of planthoppers is necessary to identify potential differences between populations.

Cixiid species such as *P. leporinus* are typically univoltine—they produce a single generation per year [4]. However, the examination of sticky traps and soil samples provided convincing evidence that the hot and dry summer in 2022 enabled *P. leporinus* to produce a second generation of adults in September, which were able to lay eggs. Under field conditions in 2020 and 2021, there was no clear evidence of bivoltine development, whereas a second generation of adults was also evident, from the monitoring of sticky traps, in 2018 [28]. The ability to produce two generations in 2022 is likely to increase the number of adults in the region in 2023. This enhances the threat of economic damage caused by *P. leporinus*, in regions where potato and sugar beet are cultivated in the same rotation. Rising temperatures, caused by climate change, are predicted to reduce crop yields, by encouraging the growth of pest insect populations [29], and projections in staple cereals such as rice, maize, and wheat indicate a yield loss of up to 25% per degree Celsius increase. Our data show a real-time change in voltinism in *P. leporinus,* probably due to the increase in temperatures.

During our field study in September, we found one *Cixius* sp. nymph on sugar beet in Eich (Rhineland-Palatinate). *Cixius wagneri* (China) is known to transmit *Candidatus* Phlomobacter fragariae, which causes marginal chlorosis in strawberries [30]. In France, *C. wagneri* in heavily-infested sugar beets, were shown to carry and transmit Arsenophonus [10]. We found one *Cixius* sp. nymph on sugar beet, but no adults in the sticky traps. Further studies monitoring cixiid species in sugar beet and potato crops, should also include *Cixius* species as potential Arsenophonus vectors in Southwest Germany. The recent finding of stolbur in sugar beet [6], and the observed host shift of *P. leporinus* to potato, may also influence the disease pressure in this region. In Ibersheim, two *H. obsoletus* individuals were found on sticky traps in sugar beet crops (data not shown). Further studies are needed to clarify the population density of *H. obsoletus* and identity which bacteria are carried. Interactions between sugar beet, potato, and vine plants could lead to an outbreak of stolbur in Rhineland-Palatinate’s vineyards and should therefore be investigated in a timely manner [15].

We conducted transmission assays to determine whether *P. leporinus* can vector Arsenophonus and/or stolbur to potato plants. Transmission from adults to sugar beet seedlings was previously reported [10,12], and was possible within an IAP of 3 days and occurred over a period of 33 days. Transmission assays from nymphs to sugar beet seedlings were also conducted previously for an IAP of 30 days, revealing a transmission frequency of 60%, based on PCR detection [12]. We provided quantitative data confirming that *P. leporinus* nymphs were able to transmit Arsenophonus to the stem, tender roots, and tubers of potato plants, after 66 days, as well as the first evidence of stolbur transmission from *P. leporinus* nymphs to sugar beet. Stolbur depends on horizontal transmission via insect vectors [31,32]. *H. obsoletus* was previously used to confirm the transmission of stolbur to sugar beet seedlings, because no stolbur-infected specimens of *P. leporinus* were available at that time [10].

We detected Arsenophonus in *P. leporinus,* as well as sugar beet root tips and potato tubers from the field. We found that 90% of all potato and sugar beet samples tested positive for Arsenophonus. The qRT-PCR results were confirmed by endpoint PCR, using the primer pairs Fra5/rP1 and Alb1/Oliv1 [3]. All potato, sugar beet, and planthopper samples showed the same band pattern. Furthermore, Arsenophonus 16S ITS sequences from different samples were identical, showing that all isolated strains shared a close genetic relationship. Further sequencing of these strains should be carried out, to determine if there is a genetic basis for the host shift from sugar beet to potato. 

In our study, only rubbery potato tubers and yellowing sugar beets were collected for analysis. Stolbur was detected in 20% of potato samples, significantly lower than the prevalence of Arsenophonus. The symptoms of Arsenophonus in sugar beet and strawberry plants are convergent with those of phytoplasmas [5]. We conclude that stolbur-like symptoms in potato can also be triggered by Arsenophonus, and 80% of the examined potato tubers were infected with Arsenophonus alone. 

More studies examining the relationship between Arsenophonus, potato plants, and planthopper vectors are urgently needed, because we currently lack an in-depth characterization of symptoms and the quantification of damage caused by these pathogens, under controlled conditions. It is also unclear where, inside the plant, Arsenophonus localizes, accumulates, and propagates. The prevalence of both pathogens in different regions of Germany is still unknown and must be assessed to facilitate appropriate containment measures. Seed potatoes could potentially spread both diseases [33]. Potential control strategies could include population suppression in *P. leporinus* and the modification of crop rotations to avoid consecutive crops of potato and sugar beet.

## 5. Conclusions

Sugar beet and potato are important root crops, with a significant socioeconomic value in the Rhine Valley. In Rhineland-Palatinate, both crops together covered more than 23,000 ha (6.0% of total arable land) in 2022 [34]. In this study, we present evidence that the planthopper *P. leporinus* has expanded its host range to potato, which will help in the development of control strategies to prevent the loss of sugar beet and potato crops. Phylogenetic analysis suggested that the planthoppers found in potato and sugar beet from different locations in Southwest Germany had a common origin. Most of the potato plants with disease symptoms such as side shoots, wilting, and rubbery tubers were colonized by Arsenophonus, in the absence of stolbur. We confirmed the ability of *P. leporinus* nymphs to transmit Arsenophonus to potato and stolbur to sugar beet. Our field data also provide evidence that *P. leporinus* can produce a second generation per year, under favorable conditions.

## Figures and Tables

**Figure 1 insects-14-00281-f001:**
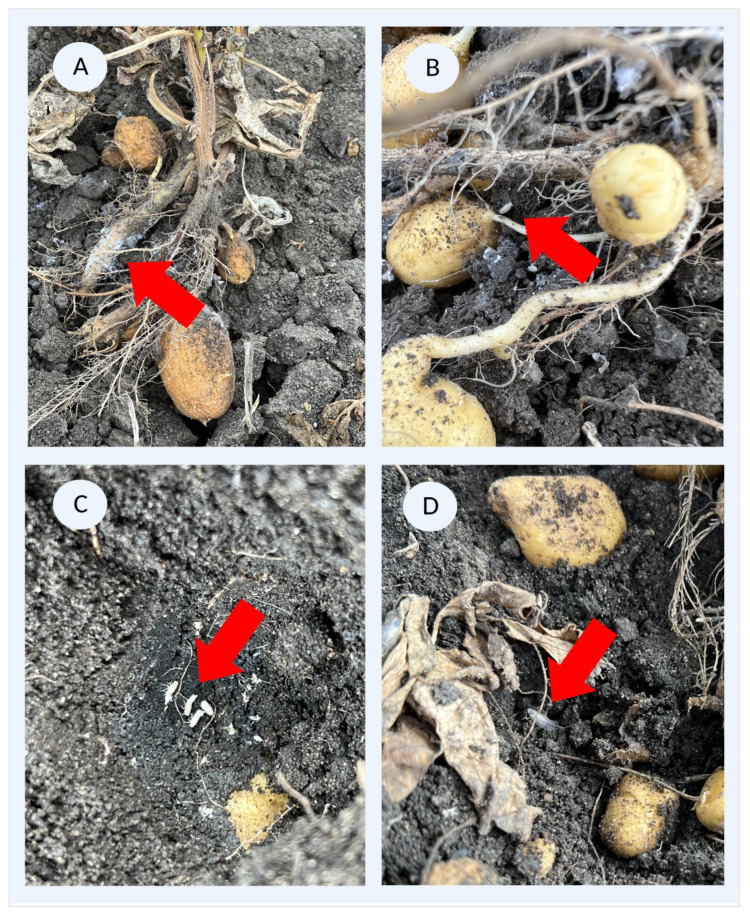
*Pentastiridius leporinus* found on potato plants in Ibersheim, September 2022. (**A**) Subterranean colonization of roots and tubers by nymphs, characterized by white, waxy filamentous residues. (**B**) Nymphs on the potato tubers and roots. (**C**) Colony of nymphs at different developmental stages and residues of molting, hidden under potato tubers. (**D**) Adult *P. leporinus* found in the soil. Red arrows indicate nymphs (**A**–**C**) and adults (**D**).

**Figure 2 insects-14-00281-f002:**
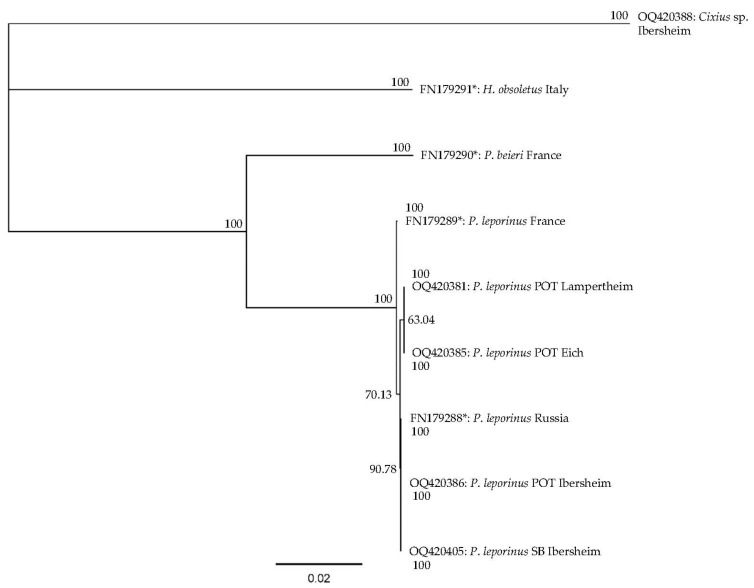
Phylogenetic relationship between cytochrome oxidase subunits I and II (COI and COII) sequences from collected *Pentastiridius leporinus*, *Hyalesthes obsoletus,* and *Cixius* sp. Specimens, and sequences of planthoppers from earlier studies (marked with *) [8]. The tree was constructed using the neighbor-joining method, based on a MUSCLE alignment of COI and COII sequences. The bar indicates the number of substitutions per site. Bootstrap values (10,000 replications) are included as numbers at the branches. POT: found in potato, SB: found in sugar beet.

**Figure 3 insects-14-00281-f003:**
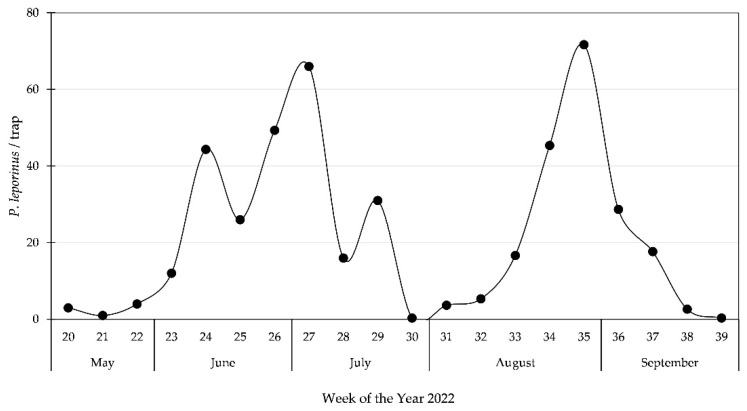
Abundance of adult *Pentastiridius leporinus* in Bickenbach, Hesse, 2022. The numbers represent means of three sticky traps per week.

**Figure 4 insects-14-00281-f004:**
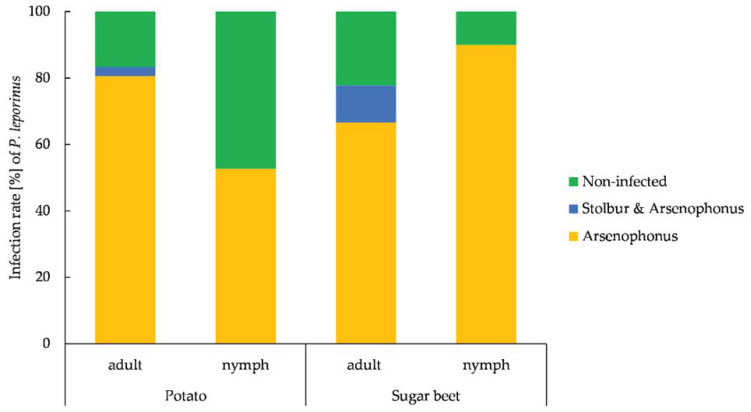
Prevalence of *Candidatus* Arsenophonus phytopathogenicus (Arsenophonus) and *Candidatus Phytoplasma solani* (stolbur) in *Pentastiridius leporinus* adults (n = 36) and nymphs (n = 36) on potato, and adults (n = 9) and nymphs (n = 10) on sugar beet plants, as determined by qRT-PCR. Nymphs and adults were collected in September 2022.

**Figure 5 insects-14-00281-f005:**
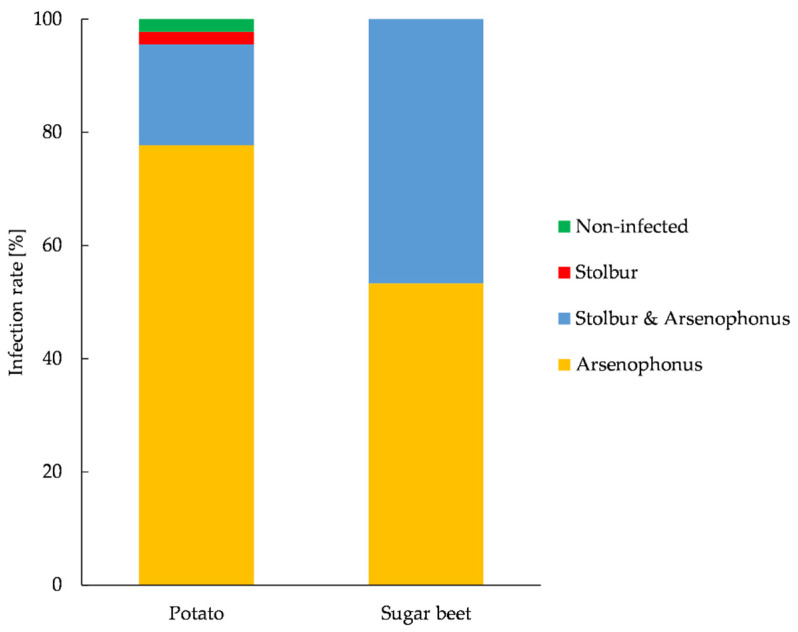
Prevalence of *Candidatus* Arsenophonus phytopathogenicus (Arsenophonus) and *Candidatus Phytoplasma solani* (stolbur) in symptomatic potato tubers (n = 45) and sugar beet roots (n = 30), in September 2022.

**Figure 6 insects-14-00281-f006:**
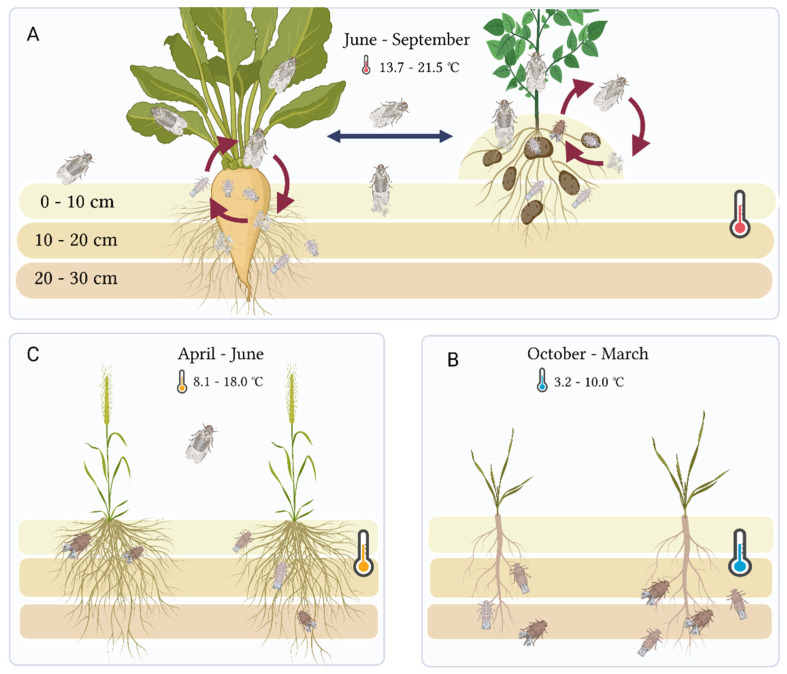
The life cycle of *P. leporinus* with two different host plants, including data [6]. (**A**) *P. leporinus* adults lay eggs, which give rise to nymphs, on sugar beet and potato plants between June and September. Under favorable conditions, a second generation of adults develops in the summer, depicted by red arrows. The blue arrow demonstrates the ability of adults to move between host plants. (**B**) Nymph distribution and temperature-dependent movement between October and March. After harvesting, the subsequent crop is usually winter wheat. (**C**) Nymph distribution and temperature-dependent movement between April and June. Thermometer symbols indicate minimum and maximum temperatures.

**Table 1 insects-14-00281-t001:** Sampling locations.

Location	State	Crops	Coordinates
Eich	Rhineland-Palatinate	Potato, sugar beet	49.765428, 8.393707
Ibersheim	Rhineland-Palatinate	Potato, sugar beet	49.726559, 8.393205
Lampertheim	Hesse	Potato	49.610846, 8.465391
Bickenbach	Hesse	Sugar beet	49.758758, 8.577285

**Table 2 insects-14-00281-t002:** Detection of *Candidatus* Arsenophonus phytopathogenicus (Arsenophonus) and *Candidatus Phytoplasma solani* (stolbur) in *P. leporinus* nymphs, as well as potato and sugar beet plants, after inoculation with field-collected third-to-fifth-instar nymphs for 66 days. Control were potato plants without nymphs.

	Plants		Group of Nymphs	
Plant Species	Arsenophonus	Stolbur	Arsenophonus	Stolbur
Potato	3/12	0/12	12/12	0/12
Sugar beet	1/3	1/3	3/3	1/3
Potato (Control)	0/3	0/3	-	-

## Data Availability

The data presented in this study are available on request from the corresponding author.

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
