# Peer review of "Potato (Solanum tuberosum) as a New Host for Pentastiridius leporinus (Hemiptera: Cixiidae) and Candidatus Arsenophonus Phytopathogenicus"

_insects, 2023, doi:10.3390/insects14030281_

Round 1

Reviewer 1 Report

Manuscript entitled "Potato (Solanum tuberosum) as a new host for Pentastiridius leporinus (Hemiptera: Cixiidae) and Candidatus Arsenophonus phytopathogenicus”

This study investigates the detection and incidence of two bacterial phytopathogens in sugar beet and potato fields in west Germany. This work enlights the importance of Pentastiridius leporinus in potato commercial fields and would be of interest to rationally managing populations.

Below are some more specific comments:

Simple summary

Line 16: The authors identified P. leporinus by morphology and “DNA analysis”..…

This sentence could be more informative because the identification was performed by morphology and comparisons of COI and COII partial sequences.

Abstract

Line 29: Why do you not mention what molecular markers have been used? Be more straightforward.

Line 34: What do you mean by “long summer”?

The summer extends from the June solstice to the September equinox, which could not be longer than that.

Figure 1. The tip of the arrows is far from the intended target.

Line 391: 6.0% of what?

Author Response

Reviewer 1

This study investigates the detection and incidence of two bacterial phytopathogens in sugar beet and potato fields in west Germany. This work enlights the importance of Pentastiridius leporinus in potato commercial fields and would be of interest to rationally managing populations.

We kindly thank reviewer 1 for his support.

Below are some more specific comments:

Simple summary

Line 16: The authors identified P. leporinus by morphology and “DNA analysis”..…

This sentence could be more informative because the identification was performed by morphology and comparisons of COI and COII partial sequences.

We thank Reviewer 1 for the suggestion and changed the sentence accordingly.

Abstract

Line 29: Why do you not mention what molecular markers have been used? Be more straightforward.

We added the information as requested.

Line 34: What do you mean by “long summer”?

The summer extends from the June solstice to the September equinox, which could not be longer than that.

By “long summer” we referred to the duration of the flight period of P. leporinus, which was longer than the years before. But we see your point that this is not self-explaining here and deleted the phrase “long”.

Figure 1. The tip of the arrows is far from the intended target.

 We thank Reviewer 1 for the suggestion and edited the figure accordingly. We changed the order of A and B, since this increases to our opinion the clarity of the figure starting with the waxy filaments.

Line 391: 6.0% of what?

We thank Reviewer 1 for the suggestion and added the information.

Reviewer 2 Report

This is a very interesting manuscript on the planthopper transmission of bacterial pathogens in potato. A very few edits are indicated in the attached pdf, after which and a minor english revision, we think that the manuscript will be ready for acceptance.

Author Response

Reviewer 2

This is a very interesting manuscript on the planthopper transmission of bacterial pathogens in potato. A very few edits are indicated in the attached pdf, after which and a minor english revision, we think that the manuscript will be ready for acceptance.

We are thankful for Reviewer 2’s support. We implemented all the suggestions for improvement made by reviewer 2 in the pdf-document.

Line 26: These bacteria cause

We changed as suggested.

Line 27: yellowings, deformed leaves and low beet yields or yellow and deformed leaves, and low beet yields

We changed as suggested.

Use authorities to all the species (insects and plants), mentioned for the first time in the main text

We thank Reviewer 2 for the suggestion and added the information.

Line 52: … and sometimes until…

We thank Reviewer 2 for the suggestion and changed the sentence accordingly.

Line 67: … both …

We thank Reviewer 2 for the suggestion and changed the sentence accordingly.

Line 69: … and…

We thank Reviewer 2 for the suggestion and changed the sentence accordingly.

Line 40: … respectively…

We thank Reviewer 2 for the suggestion and changed the sentence accordingly.

Line 84 – 88: We suggest the authors to provide info about the transmission of Stolbur prior to the specific planthopper.

We thank Reviewer 2 for the suggestion and added the information.

Line 92: We suggest the authors to use a specific voice, passive or active, throughout the M&Ms.

We thank Reviewer 2 for the suggestion and edited the chapter accordingly.

Line 97-98: Did you use a growth chamber? cage? Any special conditions held?? Please clarify. Same condition chambers in transmission assays with nymphs?

We thank Reviewer 2 for the suggestion and added the information.

Line 112: Was the fragment only one?? 1055bp?? In reference 8, they amplify a fragment of 800bp for COI and 550 bp for COII. and the primer pair here you say you use, was used in referce 8 for COII. Did you have a fragment bigger with the same primers, including both COI and COII??

The primer pair we used originates from Simon et al. 2006 (doi: https://doi.org/10.1146/annurev.ecolsys.37.091305.110018 ) code-named Dick and Barbara. It amplifies a portion of COI, a portion from gene COII. The amplification we obtained from those primer pairs was trimmed to a 972 bp long fragment, which we aligned into a tree to demonstrate phylogenetic relationship to other cixiid species. We clarified the paragraph. (Line 114 – 134)

Line 116: In total 17 adults were analyzed, for potato four from Eich, seven from ........

We thank Reviewer 2 for the suggestion and changed the sentence accordingly.

Line 118: As for the nymphs, 9 was analyzed, one from potato in Eich, two in ibersheim, four in Lampertheim and for sugar beet.....

We thank Reviewer 2 for the suggestion and changed the sentence accordingly.

Line 141: …collected nymphs (third to fifth instar)…

We thank Reviewer 2 for the suggestion and changed the sentence accordingly.

Line 183 + Line 184: …two locations… three locations…

We are sorry if we misinterpret your comment, but we assume that you are wondering why we took sugar beet samples from two and potato samples from three locations? That is because, we only found potatoes neighbored by sugar beets in two locations. We were taking samples directly after the discovery in September and during harvesting time so we had to hurry and limited knowledge.

Figure 2: it would be good to indicated with an asterisk * , the sequences used from earlier studies

We thank Reviewer 2 for the suggestion and edited the figure accordingly.

Line 437, 480 & 488. … bold…

We thank Reviewer 2 for the suggestion and edited the references accordingly.

Reviewer 3 Report

The work is devoted to the identification of the planthopper  Pentastiridius leporinus on potato and the transmission of phyotpathogens among beets and potatoes. The research is interesting for the audience working in this field and certainly has novelty. However I see a number of issues regarding the presentation of results.

Major points

1. The authors did not work in detail with the morphology of the insect at the level of different types of microscopy. Only the total length of the body and the width of the head capsule were measured. Therefore, I propose to remove information about morphology from a simple summary and discussion. I also propose to remove or replace the information about morphometry (Section 3.1.) to electronic supplementary.

2. Section 3.6. is very hard to read. This text lists data on how many insects (nymphs or adults) or plants (potatoes or beets) were infected with phytopathogens in absolute numbers or percentages. I propose instead of these enumerations to indicate the main significant effects and trends. Statistical methods should be applied to these results. I think Fisher's Exact Test would be appropriate here, as there are limits for chi square (must be > 30 observations in each column for chi square).

Other points

L 16 remove morphometry

L 59, 62, 68, 70 Missing references

L 123 Correct citation style

Figure 2. Bootstrap support of branches should be specified. Indicate on the plot the groups that authors analyzed and the groups taken from the Genbank (for example, bold or regular font).

Fig 3. I would recommend leaving the experimental points but making an approximate line (smooth graphs). Please remove the outer frame.

Table 4. Specify what control is. Tables and figures should be understandable without reference to the text.

L161-165 Is there a photo of gel electrophoresis? It should be in the main text or at least in the electronic supplementary.

Fig 4 Remove the outer frame. The legend should indicate not the total number of insects, but their number for each variant (column). Statistics are needed.

Fig 5 Same as for Fig 4

L 300. Remove morphometry

L 355 [11] ..... This is not good idea to start sentences, please rephrase.

Author Response

Reviewer 3

The work is devoted to the identification of the planthopper  Pentastiridius leporinus on potato and the transmission of phyotpathogens among beets and potatoes. The research is interesting for the audience working in this field and certainly has novelty. However I see a number of issues regarding the presentation of results.

We kindly thank reviewer 1 for his suggestions for improvement.

Major points

  1. The authors did not work in detail with the morphology of the insect at the level of different types of microscopy. Only the total length of the body and the width of the head capsule were measured. Therefore, I propose to remove information about morphology from a simple summary and discussion. I also propose to remove or replace the information about morphometry (Section 3.1.) to electronic supplementary.

We followed the suggestion of Reviewer 3 and removed the information from the simple summary and discussion. We moved the morphometric information to the supplementary File (Table S1).

  1. Section 3.6. is very hard to read. This text lists data on how many insects (nymphs or adults) or plants (potatoes or beets) were infected with phytopathogens in absolute numbers or percentages. I propose instead of these enumerations to indicate the main significant effects and trends. Statistical methods should be applied to these results. I think Fisher's Exact Test would be appropriate here, as there are limits for chi square (must be > 30 observations in each column for chi square).

We kindly disagree on this point. In section 3.6 the prevalence of Arsenophonus and Stolbur from field samples were presented as found, so we do not see the necessity to perform statistical analysis on our reported findings.

Other points

L 16 remove morphometry

We changed the manuscript accordingly.

L 59, 62, 68, 70 Missing references

We thank Reviewer 3 for the suggestion and added the information.

L 123 Correct citation style

We thank Reviewer 3 for the suggestion and added the information.

Figure 2. Bootstrap support of branches should be specified. Indicate on the plot the groups that authors analyzed and the groups taken from the Genbank (for example, bold or regular font).

We changed the figure accordingly.

Fig 3. I would recommend leaving the experimental points but making an approximate line (smooth graphs). Please remove the outer frame.

We edited the figure accordingly.

Table 4. Specify what control is. Tables and figures should be understandable without reference to the text.

We changed the table accordingly.

L161-165 Is there a photo of gel electrophoresis? It should be in the main text or at least in the electronic supplementary.

We added a photo of gel electrophoresis as supplementary figure S1. The text reference can be found in Line 269.

Fig 4 Remove the outer frame. The legend should indicate not the total number of insects, but their number for each variant (column). Statistics are needed.

We followed the suggestion of Reviewer 3. Please see our comment above on statistics.

Fig 5 Same as for Fig 4

We followed the suggestion of Reviewer 3. Please see our comment above on statistics.

L 300. Remove morphometry

We changed the manuscript accordingly.

L 355 [11] ..... This is not good idea to start sentences, please rephrase.

We changed the sentence accordingly.

Round 2

Reviewer 3 Report

Most of my comments have been reflected . The authors do not agree with one of the major points, but I leave it to their choice.